# Cutting Movement Assessment Scores during Anticipated and Unanticipated 90-Degree Sidestep Cutting Manoeuvres within Female Professional Footballers

**DOI:** 10.3390/sports10090128

**Published:** 2022-08-24

**Authors:** Chloe Needham, Lee Herrington

**Affiliations:** 1Centre for Health Sciences Research, University of Salford, Salford M6 6PU, UK; 2Birmingham City Football Club, Wast Hills Training Ground, Birmingham B38 9EL, UK

**Keywords:** injury screening, anterior cruciate ligament, cutting task

## Abstract

**Background:** ACL injuries present a considerable burden in female football, with highest incidence being related to change of direction (COD) tasks. The aim was to identify if differences existed between an anticipated and unanticipated 90-degree cutting task using the CMAS. **Methods:** 11 female professional footballers completed twelve 90-degree COD tasks (6 anticipated, 6 unanticipated). Participants performed the unanticipated task in response to a moving football at the start of their acceleration. All COD tasks were filmed and assessed using the CMAS. **Results:** The CMAS score for the unanticipated COD task (5.53 ± 0.71) was significantly larger than for the anticipated COD task (3.55 ± 0.85, *p* < 0.012). Excellent intra-rater reliability was observed (ICC = 0.97) for analysis of CMAS scores. **Conclusions:** Female footballers in this sample demonstrated a greater CMAS score during an unanticipated COD task compared to an anticipated COD task. These athletes are therefore more likely to display ‘high-risk’ movement patterns, thus greater risk of injury. Reacting to a sporting implement, such as a moving ball, may be a contributing factor to these results. Further research into unanticipated COD tasks should be considered to determine why these differences occur and the impact of anticipation on performance.

## 1. Introduction

Anterior cruciate ligament (ACL) injuries are a significant musculoskeletal injury in multi-directional team sports [1,2], with an annual incidence rate of 68.6 per 100,000 person-years [3], and football demonstrates the highest prevalence of ACL injuries across sport [4]. ACL injuries are detrimental to the athlete, with associated financial costs [5], long-term altered gait patterns [6], and high-risk of developing osteoarthritis [7]. In addition, lengthy periods of rehabilitation following surgery [8] can result in entire seasons of missed playing time, which can a significant impact both on individual and team performance.

The majority of ACL injuries are non-contact injuries, occurring when landing from a jump [2,9], or changing direction through sidestep cutting [10,11]. Sidestep cutting accounts for the largest proportion of change of direction (COD) actions within a football match, where athletes are reported to change direction less than 90º over 250 times during a single game [12,13]. These movements have the potential to generate high knee joint loads, through placing the lower limb in a position of ‘dynamic valgus’, described as knee abduction combined with hip adduction and internal rotation [14,15]. In addition, a combination of ‘high-risk’ movement patterns including foot plant distance, internal hip rotation, knee flexion angle, and trunk rotational angles [10,16,17], often results in increased knee abduction moments (KAMs) increasing the load on the ACL.

Screening tools such as 3D motion analysis have been described as the gold standard method for assessing movement [18] and have recognise athletes at risk of ACL injury through identification of KAMs [14]. Despite this, the use of 3D motion analysis has limitations around accessibility and cost, steering the development of more practical ‘user friendly’ qualitative screening tools. Qualitative screening typically requires assessment of athletes performing a movement pattern such as a squat or landing task against a scoring system. For example, The Qualitative Assessment of Single Leg Loading (QASLS) and Landing Error Scoring System (LESS) have been shown to identify risk factors associated with lower limb injury [19,20]. Both screening tools assess vertically orientated tasks. Sidestep cutting remains a dominate mechanism of ACL injury [10,11], and therefore screening tools for field-based cutting are required. 

The Cutting Movement Assessment Score (CMAS) developed by Jones et al. [21], identifies movement deficits in the frontal and sagittal plane when sidestep cutting. Errors that are scored within the CMAS include trunk lean, foot width and plant angle, knee valgus position and hip rotation, all of which have been associated with higher ground reaction forces and peak KAMs [21]. Preliminary research has identified a strong relationship between CMAS score and peak KAMs (*r* = 0.63; *p* < 0.001), and moderate to excellent intra and inter-rater agreement (intra-rater: *k* = 0.60–1.00, 75–100% agreements; inter-rater: *k* = 0.71–1.00, 87.5–100% agreements). Dos’Santos et al. [22] utilised a modified version of the CMAS to identify a large significant relationship between CMAS and peak KAMs (*r* = 0.80; *p* < 0.001), validating CMAS as a screening tool against 3D motion analysis to identify athletes who display ‘high-risk’ cutting kinematics. The current research into the CMAS lacks ecological validity as the research has been conducted in a laboratory setting. In addition, the COD task is pre-planned, limiting the influence of decision-making impacting performance of the task. Team sport athletes reactively change direction in response to actions from opponents or team-mates which are often unpredictable, and therefore an anticipated task is not a true reflection of the demands within these sporting environments.

Several researchers have made efforts to increase the ecological validity of their COD task. Kristianslund et al. [23] implemented a human stimuli and catching a handball on the approach to their change of direction task. Besier et al. [24], observed several biomechanical differences between an unanticipated versus pre-planned cutting manoeuvre, determined by a random reactive sequence on an LED display. They discovered that the knee valgus moments were up to two times greater during unanticipated manoeuvres compared to pre-planned tasks (*p* < 0.05). 

The first aim of this study was to assess the intra-rater reliability of the CMAS. Then, the study aimed to identify if differences existed between an anticipated and unanticipated 90-degree cutting task using the CMAS score in female professional footballers. The final aim was to examine inter-limb differences existed in the CMAS score during a 90-degree cutting task. It was hypothesised that the unanticipated 90-degree cutting task would display a greater CMAS score and that the non-dominate leg would display a greater CMAS score for both the anticipated and unanticipated 90-degree cutting task. 

## 2. Methods 

The research was approved via an ethics review board (ethics number: HST2021-125) and all participants gave written informed consent.

The sample was recruited from a professional team competing in the top tier of women’s football in England. Eleven female players participated (mean ± SD; age: 24 ± 3.7 years, height: 1.66 ± 4.5 m, mass: 63 ± 6.1 kg, dominate foot: right n = 10 left n = 1). For inclusion, they had to be over the age of 18, had trained in a structured programme for three years or more and participated in four or more training sessions a week in addition to a competitive fixture. Participants were excluded if they had a current injury or previous significant knee injury. 

Prior to the task, the participants completed a standardised warm up to ensure readiness to complete the task. All participants completed the task on the same day, during a ‘normal’ week’s training schedule.

For the main task, a standardised layout of the COD task was followed (Figure 1a,b) [21,22]. The task took place on an indoor 3G artificial surface and the participants wore a self-selected pair of football boots. The participants performed one practise trial for both the anticipated and unanticipated task. Time to complete the task was measured using Brower TCi-Gate Timing System as a performance marker to ensure individual consistency between attempts. Gates were positioned at the start and end point to record time to complete the task in seconds.

The participant completed three trials cutting to the left and three trials to the right for both anticipated and unanticipated tasks (12 trials in total). To manage order effects and the effects of fatigue, a randomised task order was created with both direction and task nature randomised. From the starting point (first speed gate), the participants were instructed to accelerate maximally for five meters towards a flat disk (20cm in diameter) where they changed direction to the left or right and continued to accelerate three meters towards the end point (second speed gate). The two independent variables were defined as follows:

Anticipated task—the participant was given the direction of ‘left’ or ‘right’ prior to starting the task. 

Unanticipated task—prior to setting off the participant was instructed to react and cut to the left or right at the flat disk dependent upon the direction of a moving football. This ball was kicked by the same researcher towards a mini goal within a standardised area either towards the participants left or right as soon as the participant started accelerating from the starting gate. 

During the cutting task, participants were filmed simultaneously using two Akaso Brave 7 cameras, sampling at 60Hz. One camera was placed in the frontal plane and one camera in the sagittal plane, five meters away from the flat disk where the participants changed direction. The HD MP4 video files (1080 pixels) were then viewed for analysis.

Each video was assessed using the modified version of the CMAS tool [22] (Table 1). The same researcher reviewed all footage giving the participant a score out of 11 based on the errors identified on the CMAS. The data was analysed using SPSS 21.0. A mean (±SD) CMAS score was produced per participant for the left unanticipated, right unanticipated, combined unanticipated, left anticipated, right anticipated, and combined anticipated tasks. 

Reliability of each CMAS repetition per participant within each task was assessed via within subject co-efficient of variation, in addition to the individuals time to complete the task for each condition. Intra-rater reliability was determined by the same researcher viewing five randomly selected videos on two separate occasions, seven days apart. The scores from the two occasions were analysed using intraclass correlation coefficients (ICC) and the standard error of measurement (SEM) was calculated. The SEM was used to calculate the minimal detectable change (MDC) using the following formula: 1.96 × SEM × √2. The ICC was interpretated as poor (<0.50), moderate (0.50–0.75), good (0.75–0.90) and excellent (>0.90) [25,26]. 

The difference between the CMAS scores relating to task anticipation and limb differences were determined using a two-way ANOVA. A Post hoc analyses with Bonferroni Correction was carried out. To explore the magnitude of differences between groups, Hedges’ g effect size with 95% confidence intervals (CI) was calculated. The effect size was interpretated as follows; (<0.19) small, (0.20–0.59) moderate, (0.60–1.19) large, (1.20–1.99) very large, (2.0–3.99) [25,26].

## 3. Results

Excellent intra-rater reliability was observed (ICC 0.97 (95% CI from 0.74 to 1.0 (*p* ≤ 0.001), SEM = 0.5 points, SEM% = 8%, MDC = 1.4 points). The within subject coefficient of variation for each condition is displayed in Table 2.

The mean (±SD) CMAS score are displayed in Table 2. Normality of data was confirmed by Shaprio–Wilks test (*p* > 0.05). A two-way ANOVA analysed the effect of limb difference and anticipation on CMAS score. The simple main effects analysis showed that there was no statistical significance between limb and CMAS score (left; 4.8 ± 0.8 vs. right; 4.3 ± 0.7, *p* = 0.2). A second simple main effects analysis showed that there was a statistical significance between anticipation of COD task and CMAS score (*p* ≤ 0.005). Post hoc analyses with Bonferroni correction revealed that the CMAS score for the unanticipated task (5.5 ± 0.7) was significantly larger compared to the anticipated task (3.6 ± 0.9), (*p* < 0.012, effect size = 2.52).

## 4. Discussion

The aim of this study was to identify differences between an anticipated and unanticipated 90-degree cutting task using the CMAS. A higher CMAS score on the unanticipated task compared to the anticipated task occurred (5.5 ± 0.71 vs. 3.6 ± 0.9 points, *p* < 0.012). There was no significant difference between left versus right limb on the CMAS score during both 90-degrees cutting tasks (left; 4.8 ± 0.8 vs. right; 4.3 ± 0.7, *p* = 0.175).

The results demonstrated that female footballers are more likely to display a greater CMAS score during an unanticipated COD task. This suggests that during an unanticipated COD task, female footballers are likely to display more ‘high-risk’ movement patterns generating higher knee joint loads. A combination of these ‘high risk’ movement patterns increases KAMs and potentially increases the load on the ACL and overall risk of injury. With a significantly high number of sidestep cuts completed during a single football match [12,13], this may provide an explanation as to why non-contact ACL injuries are most prevalent in team sports such as football [4] or handball [15], due to the multi-directional reactive nature of the sport. The CMAS tool can be used to identify this as a ‘high-risk’ movement pattern that increases KAMs which other screening tools may not be able to recognise [7]. As the MDC value was lower than the mean difference in CMAS score between tasks (1.4 vs. 2.0), it appears that the significant increase in CAMS score was due to the change of task, rather than chance or error.

Scharfen and Memmert [27] suggested that cognitive performance is positivity related to football-specific technical skills including COD and dribbling a ball, signifying an interplay between cognitive demands and sporting performance. Due to the difference in cognitive demands during an open and closed skill [28], an additional stimulus (such as reacting to the ball) increases cognitive load and could be a contributing factor towards a greater CMAS score during the unanticipated task. The decision to add a sports stimulus in the form of a moving football was important to increase the ecological validity of the COD task. During data collection, there was potential for human error when starting the motion of the football at a consistent time for each participants acceleration. This margin of error may have altered the response time for each athlete during the unanticipated task, potentially impacting COD technique and CMAS score. Nevertheless, by responding to the movements of a football by a human researcher allowed the task to be more representative of gameday demands, where athletes change direction in response to actions from opponents. In comparison, similar research from Besier et al. [24] investigating the impact of anticipation on knee joint loads used an LED display to inform participants which way to change direction. Using an LED display eliminates human error in delivery of the stimuli but reduces overall ecological validity and thus ability to generalise the findings to a sport specific COD task. The advantages of reacting to a human stimuli or sporting implement versus the use of an electronic display to determine COD needs to be considered in future research.

Previous research noted significant differences in limb dominance [29]^,^ this study demonstrated no significant difference between limbs on the CMAS score for both tasks (left; 4.8 ± 0.8 vs. right; 4.3 ± 0.7, *p* = 0.175). With 10 out of 11 participants right foot dominant, it is unclear what the impact limb dominance is on COD performance. The dominant limb is reported to accumulate 82–84% of touches during an average football match at the World Cup [30], during which the dominate limb performs coordinated intersegmental movements to impact the ball, and the non-dominate limb is a stabilizer to provide a foundation for movement. Considering this, if the participants changed direction whilst dribbling a ball, differences may have been observed in lower limb mechanics due to the interaction between football and limbs.

Reliability amongst subjective movement screening tools is generally only ‘acceptable’ [19], due to rater blinding, small sample sizes and lack of descriptive rater information. This research demonstrated excellent intra-rater reliability (ICC = 0.97, 95% CI; 0.74–1.0), supporting Dos’Santos et al. [22] and Jones et al. [21]. In this research, a large range of coefficient of variation (21.4–42.9%) across all tasks was observed. This may be attributed to learning effect or acute fatigue, as the participated completed 12 COD tasks in total, double the number in Dos’Santos et al^.^ [22].

Timing gates provided a performance indicator regarding task completion time with consistent results between the two tasks, in addition to adding competitiveness amongst participants to complete the task in the fastest time. This also gives confidence that all trials were executed with a high intensity acceleration, and differences in CMAS score was likely not to be attributed to acceleration and speed of approach.

The participants were from the same team, participating in identical training. In comparison to previous research, Dos’Santos et al. [22]. utilised four different multi-directional sports. Multiple sports may display different COD demands, for example, the COD demands of a cricketer are predominantly 180-degrees during wicket runs, compared to a footballer who mostly changes direction between 0–90 degrees [12]. Using a single sport sample enhances understanding of the specific COD demands associated with this sporting group.

A greater sample size may have allowed more evidence to define the significance between the difference in CMAS scores. Nevertheless, as participants were l from the same team, variables such as training loads and methods were controlled. In the future, it may be best practice to seek out a variety of professional teams from the same sporting league to gain a greater sample size within the same sport.

There is limited research into COD task screening tools, and minimal on unpredictable COD tasks. Further research into the CMAS and task anticipation should consider whether any individual variables of the CMAS increase during the unanticipated task compared to the anticipated task. For example, an increase in lateral leg plant width may demonstrate the greatest increase across all CMAS variables on the unanticipated task and would indicate further research into a single movement deficit.

The research focused solely on female footballers. As the overall incidence of ACL injuries is higher in female footballers compared to males [4] it is important to explore why these differences in injury rates occur. Further research could be undertaken by replicating the study on male footballers to determine if similar findings exist.

## 5. Conclusions

Practitioners should consider assessing anticipated and unanticipated COD tasks to replicate the COD movements patterns demonstrated within team sports. To complete a comprehensive ACL injury screening profile, though, the CMAS could be paired with a landing task screening tool to ensure all types of ACL injury mechanisms are covered. The findings support the importance of including reactive COD tasks within rehabilitation following injury. From this, interventions during both anticipated and unanticipated tasks can be implemented to reduce displaying ‘high-risk’ movement patterns when COD.

Female footballers displayed a higher CMAS score during an unanticipated compared to an anticipated COD task. There was no significant difference between limbs in the CMAS scores during both tasks. From this, it can be considered that the addition of a stimuli to reactively determine cutting direction may results in an athlete displaying more ‘high-risk’ cutting mechanics. Future research into COD screening should consider the effect of task anticipation on COD technique, alongside each individual CMAS variable. Practically, this research should remind sports practitioners the importance of unanticipated reactive COD tasks within screening, rehabilitation, and performance.

## Figures and Tables

**Figure 1 sports-10-00128-f001:**
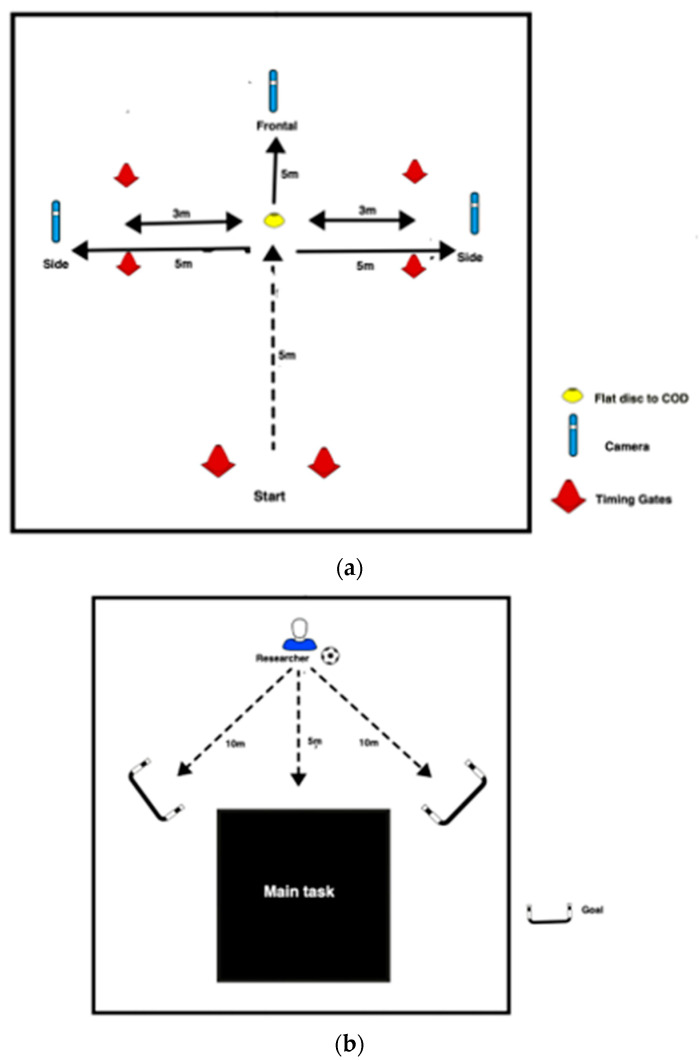
(**a**) Plan view of the standardised setup for the 90° change of direction task. (**b**) Plan view of the standardised setup for the unanticipated task. Key: COD: change of direction; m: meters.

**Table 1 sports-10-00128-t001:** Cutting movement assessment score (CMAS) Tool.

Camera Angle	Variable	Observation	Score
	**Penultimate contact**		
Side/45°	Backwards inclination of the trunk	Y/N	Y = 0/N = 1
	**Final contact**		
Front/45°	Wide lateral leg plant (at initial contact)	Y/N	Y = 2/N = 0
Front/45°	Hip in an initial internally rotated position (at initial contact)	Y/N	Y = 0/N = 1
Front/45°	Initial knee ‘valgus’ position (at initial contact)	Y/N	Y = 0/N = 1
Front/side	Inwardly rotated foot position at initial contact)	Y/N	Y = 0/N = 1
Front/45°	Frontal plank trunk position relative to intended direction; Lateral (L), Upright (U) or Medial (M) (at initial contact and over WA)	L/U/M	L = 2/U = 1/M = 0
Side/45°	Trunk upright or leaning back throughout contact (at initial contact and over WA)	Y/N	Y = 0/N = 1
Side/45°	Limited knee flexion during final contact (over WA)	Y/N	Y = 0/N = 1
Front/45°	Excessive knee ‘valgus’ motion during contact (over WA)	Y/N	Y = 0/N = 1
		Total score	/11

Key: WA: weight acceptance; Y: yes; N: no; L: lateral; U: upright; M: medial.

**Table 2 sports-10-00128-t002:** Mean, SD and CV for CMAS score and time to complete task.

	CMAS Score (/12)			Time to Complete Task (s)		
	Mean	SD	Within Subject CV	Mean	SD	CV
**Left Anticipated**	3.8	0.9	40.8%	1.8	0.1	3.9%
**Right Anticipated**	3.3	0.8	45%	1.8	0.1	3.2%
**Combined Anticipated**	3.6	0.9	42.9%	1.8	0.1	3.6%
**Left Unanticipated**	5.9	0.8	21.4%	1.8	0.1	3.2%
**Right Unanticipated**	5.2	0.7	34.3%	1.7	0.1	4.4%
**Combined Unanticipated**	5.5	0.7	27.9%	1.8	0.1	5.8%

Key: CMAS: cutting movement assessment score; SD: standard deviation; (s): seconds; CV: coefficient variation.

## Data Availability

Not applicable.

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
