# Peer review of "Cutting Movement Assessment Scores during Anticipated and Unanticipated 90-Degree Sidestep Cutting Manoeuvres within Female Professional Footballers"

_sports, 2022, doi:10.3390/sports10090128_

Round 1

Reviewer 1 Report

The manuscript is clearly written in professional, unambiguous language. There are well-developed and supported arguments that meet the goals set out in the Introduction. The research questions are well defined and authors have conducted necessary experiments to support their study.

1. The novelty and originality of this research needs to be more fully demonstrated. Because in general, compared to anticipated situations, unanticipated 90-degree sidestep cutting manoeuvres forces the athlete to react hastily without adequate preparation, resulting in movement distortion. This conclusion is easy to know by reasoning through common sense.

2. The font size on figure 1a and 1b is too small to read some text.

3. When using the CMAS tool, the same researcher reviewed all footage and gave the participant a score. Given the effects of subjectivity, is it possible that different researchers will get different scores when grading?

4. Table 2 illustrates the times to complete tasks in different situations. But the manuscript does not explain the usefulness of recording this indicator in the experiment

Author Response

The manuscript is clearly written in professional, unambiguous language. There are well-developed and supported arguments that meet the goals set out in the Introduction. The research questions are well defined and authors have conducted necessary experiments to support their study.

  1. The novelty and originality of this research needs to be more fully demonstrated. Because in general, compared to anticipated situations, unanticipated 90-degree sidestep cutting manoeuvres forces the athlete to react hastily without adequate preparation, resulting in movement distortion. This conclusion is easy to know by reasoning through common sense.

 We would respectfully disagree, though apparently obvious this has not been previously shown in the literature, so makes the study worthwhile and impactful

  1. The font size on figure 1a and 1b is too small to read some text.

 This has been amended

  1. When using the CMAS tool, the same researcher reviewed all footage and gave the participant a score. Given the effects of subjectivity, is it possible that different researchers will get different scores when grading?

 Previous research by DosSantos, has shown this is the case, but as we didn’t use multiple testing, and reported the reliability of the tester involved, we fail to see the relevance for this paper

  1. Table 2 illustrates the times to complete tasks in different situations. But the manuscript does not explain the usefulness of recording this indicator in the experiment

This has been expanded upon in the discussion

Reviewer 2 Report

Dear authors, 

I would like to congratulate you on your work. The ecological approach to this problem is very welcome in this reviewer's opinion. I have carefully analysed the manuscript and I suggest you to consider the following recommendations:

Please add information about the training sessions (frequency and/or duration x week), and number of matches (during a typical season) played by the participants.

Within Methods section I suggest you to add information about the total number of tasks completed by each player (e.g., the participant completed 12 trials overall, three trials cutting to the left and three trials to the right for both anticipated and unanticipated tasks) as well as the recovery time between trials. Moreover, I suggest you to explain better the randomisation strategy [i.e., only direction (left and right) or both direction (left/right) as well as task (anticipated/unanticipated) were randomised?]

Please report the video file resolution (Pixel width x Pixel height or "HD", "Ultra HD", ecc.) and format (MP4, MOV, AVI) you have used to store it, as well as the video player software and the features of the computer you have used to play the videos. 

Within the Results as well as the Discussion sections you report the differences between Left and Right limb but it is not consistent with your aims. In particular, you were focused on differences between dominant and non-dominant leg, which were not the same for all the participants (dominate foot: right n=10 left n=1). Therefore, dominant vs non-dominant terms should be used instead of left vs right.

Author Response

I would like to congratulate you on your work. The ecological approach to this problem is very welcome in this reviewer's opinion. I have carefully analysed the manuscript and I suggest you to consider the following recommendations:

Please add information about the training sessions (frequency and/or duration x week), and number of matches (during a typical season) played by the participants.

This has now been included

Within Methods section I suggest you to add information about the total number of tasks completed by each player (e.g., the participant completed 12 trials overall, three trials cutting to the left and three trials to the right for both anticipated and unanticipated tasks) as well as the recovery time between trials.

This has now been included

Moreover, I suggest you to explain better the randomisation strategy [i.e., only direction (left and right) or both direction (left/right) as well as task (anticipated/unanticipated) were randomised?]

This has now been included

Please report the video file resolution (Pixel width x Pixel height or "HD", "Ultra HD", ecc.) and format (MP4, MOV, AVI) you have used to store it, as well as the video player software and the features of the computer you have used to play the videos. 

This has now been included

Within the Results as well as the Discussion sections you report the differences between Left and Right limb but it is not consistent with your aims. In particular, you were focused on differences between dominant and non-dominant leg, which were not the same for all the participants (dominate foot: right n=10 left n=1). Therefore, dominant vs non-dominant terms should be used instead of left vs right.

As no significant difference was found between limbs, we kept the left and right description for ease

Round 2

Reviewer 1 Report

I think this article is ready for publication.